# The Dimeric Form of HPV16 E6 Is Crucial to Drive YAP/TAZ Upregulation through the Targeting of hScrib

**DOI:** 10.3390/cancers13164083

**Published:** 2021-08-13

**Authors:** Lorenzo Messa, Marta Celegato, Chiara Bertagnin, Beatrice Mercorelli, Gualtiero Alvisi, Lawrence Banks, Giorgio Palù, Arianna Loregian

**Affiliations:** 1Department of Molecular Medicine, University of Padua, 35121 Padua, Italy; lorenzo.messa@unipd.it (L.M.); marta.celegato@unipd.it (M.C.); chiara.bertagnin@unipd.it (C.B.); beatrice.mercorelli@unipd.it (B.M.); gualtiero.alvisi@unipd.it (G.A.); giorgio.palu@unipd.it (G.P.); 2International Centre for Genetic Engineering and Biotechnology, 34149 Trieste, Italy; banks@icgeb.org

**Keywords:** cell polarity, HPV16 E6, Scribble module, YAP/TAZ

## Abstract

**Simple Summary:**

Understanding the mechanisms of action of HPV oncoproteins is pivotal for the rationale development of anti-cancer drugs to treat HPV-related malignancies. The aim of the present study was to explore more in detail the mechanism of action of the HPV16 oncoprotein E6 that directly fosters the YAP/TAZ signaling pathway, a conserved cascade highly active in HPV-related cancers. We confirmed previous evidence about the importance of the PDZ-protein targeting in this process, highlighting here the importance of hScrib degradation, and discovered that the targeting of the Scribble module involves the dimeric form of HPV16 E6. The findings here presented extend our knowledge about the mechanism through which the oncoprotein E6 targets a PDZ-host factor to degradation in cancer cells.

**Abstract:**

Human papillomavirus is the most common viral infectious agent responsible for cancer development in humans. High-risk strains are known to induce cancer through the expression of the viral oncogenes *E6* and *E7*, yet we have only a partial understanding of the precise mechanisms of action of these viral proteins. Here we investigated the molecular mechanism through which the oncoprotein E6 alters the Hippo-YAP/TAZ pathway to trigger YAP/TAZ induction in cancer cells. By employing E6 overexpression systems combined with protein–protein interaction studies and loss-of-function approaches, we discovered that the E6-mediated targeting of hScrib, which supports YAP/TAZ upregulation, intimately requires E6 homodimerization. We show that the self-association of E6, previously reported only in vitro, takes place in the cytoplasm and, as a dimer, E6 targets the fraction of hScrib at the cell cortex for proteasomal degradation. Thus, E6 homodimerization emerges as an important event in the mechanism of E6-mediated hScrib targeting to sustain downstream YAP/TAZ upregulation, unraveling for the first time the key role of E6 homodimerization in the context of its transforming functions and thus paving the way for the possible development of E6 dimerization inhibitors.

## 1. Introduction

Human papillomavirus (HPV) is the most common cancer-causing virus worldwide [1]. Although humans can be infected by more than 200 different viral strains, only a small group can drive tumor development [2]. These so-called “high-risk” genotypes induce cancer as a consequence of a persistent infection at specific anatomical sites, causing virtually all cases of cervical cancer and an increasing fraction of anogenital and oropharyngeal tumors [2,3,4]. Despite substantial advancement in HPV prophylaxis for cancer prevention, the virus is still responsible for approximately 5% of all human malignancies, with more than 500,000 new cervical cancer cases documented in 2018 [5,6]. Unfortunately, no specific anti-HPV drugs exist in the clinics for the treatment of HPV-associated cancers. At the basis of HPV-induced cancer development lie two viral oncoproteins, E6 and E7, whose functions are required by the virus to maintain a basal-like state in the infected keratinocyte to sustain viral replication [7]. Their synergistic activities profoundly perturb the cellular environment and can thus predispose the cell to malignant transformation [8]. E6 and E7 work solely through protein–protein interactions (PPIs), and the perturbation of cell homeostasis is mainly achieved by hijacking the cellular proteasome machinery to induce the proteasomal degradation of several host factors, including p53 and pRb, respectively [9].

In the last decade, many studies on the biology of solid cancers, including HPV-related tumors, addressed the importance of the Hippo-YAP/TAZ cascade. YAP/TAZ signaling has emerged as a core signaling pathway in malignant lesions dictating a wide range of pro-tumorigenic responses (reviewed in [10]). YAP and TAZ are transcriptional co-activators that promote the transcription of several target genes involved in multiple biological phenomena, among which the acquisition of stem-like traits has emerged as a hallmark of YAP/TAZ activity [11]. Indeed, YAP and TAZ have been shown to exert a central role in the determination of somatic stem cells and downstream stem cell-related functions in adult healthy tissues [12]. In solid tumors, cancer cells exhibit a profound addiction to YAP and TAZ for cell survival, proliferation, apoptosis-resistance, and stemness, and sustained YAP/TAZ activity is a well-recognized determinant of cancer-stem cell populations [13].

In the context of HPV-driven tumorigenesis, recent studies have shown that both E6 and E7 are independently capable of sustaining the activation of YAP/TAZ in different ways, either directly, by modulating the fate of proteins that regulate YAP/TAZ function, or indirectly, by altering signaling cascades that converge on Hippo-YAP/TAZ regulation (reviewed in [14]). Understanding the precise contribution of E6 and E7 in promoting YAP and TAZ could reveal new attractive targets for the development of therapeutics against HPV-induced cancers. So far, the main documented activity of E7 that directly promotes YAP/TAZ signaling is the degradation of the non-receptor tyrosine phosphatase PTPN14 [15,16], whose role in the Hippo pathway has been previously reported [17]. In contrast, the specific activity(ies) of the oncoprotein E6 that directly sustains YAP/TAZ induction in cancer cells is only partially understood. In this context, the E6-mediated degradation of both p53 and some PDZ domain-containing proteins (termed hereafter as PDZ proteins) represent possible upstream events responsible for direct YAP/TAZ upregulation and activation [18,19,20,21]. Indeed, it was recently shown that PDZ protein-degradation mediated by E6 contributes to YAP nuclear retention [22].

In the present study we investigated the molecular mechanism through which E6 of HPV16, the most common mucosal high-risk HPV type, promotes YAP/TAZ upregulation by targeting the PDZ-protein hScrib. We present evidence that E6 homodimerization is an event that is important for the E6-mediated targeting of hScrib and instrumental in promoting downstream YAP/TAZ induction. Mechanistically, E6 dimerizes in the cytosol and, as a dimer, forms a protein complex with hScrib to target hScrib for proteasomal degradation. Our findings dissect the molecular mechanisms behind the E6-induced hijacking of the Scribble module and highlight the importance of E6 self-association for its transforming activities.

## 2. Materials and Methods

### 2.1. Cell Lines

In the present study we solely used epithelial cells. The HPV-negative epithelial cell lines used were: HEK 293T (transformed human embryonic kidney cells), HaCaT (non-transformed immortalized human skin keratinocytes), HEKn (human epidermal keratinocytes), C33A (human cervical cancer cells), and H1299 (human lung non-small cancer cells). The HPV16-positive cell lines used were: CaSki (human cervical cancer cells) and SiHa (human cervical cancer cells). All cell lines were purchased from ATCC (Manassas, VA, USA) except for HaCaT cells which were a kind gift form Prof. Enzo Di Iorio (University of Padua, Italy) and HEKn which were purchased from Sigma-Aldrich (St. Louis, MO, USA). All cell lines were cultured in Dulbecco’s Modified Eagle’s medium (DMEM, Thermo Fisher Scientific, Waltham, MA, USA) supplemented with 10% fetal bovine serum (FBS, Thermo Fisher Scientific, Waltham, MA, USA) except for HEKn which were cultured in Keratinocyte Growth Medium (Sigma-Aldrich, St. Louis, MO, USA). Cells were grown at 37 °C with 5% CO_2_ and were tested monthly for mycoplasma contaminations. We did not regularly use Pen/Strep in cell culture media.

### 2.2. Plasmid Construction and Mutagenesis

The pcDNA3-RLuc, pcDNA3-RLuc-E6, and pcDNA3-RLuc-p53 plasmids have previously been reported [23]. Experiments involving the expression of untagged E6 proteins were performed with p513-empty vector, p513-E6, p513-E6 Y43E/F47R, and p513-E6 T149D/L151A plasmids. The latter was constructed by transferring the E6 T149D/L151A-encoding sequence from the GW1-HA16E6-T149D/L151A plasmid [24] to the p513-empty vector with BamHI and NotI restriction enzymes (NE BioLabs, Ipswich, MA, USA), while the others were previously reported [23]. The pDEST6.2-YFP-E6 plasmid was constructed by cloning the HPV16 E6 sequence from the p513-E6 vector into the pDEST6.2-nYFP plasmid with Gateway^®^ cloning technology (Thermo Fisher Scientific, Waltham, MA, USA) as previously described [23]. The pDEST6.2-YFP-E6_N_ and pcDNA3-RLuc-E6_N_ plasmids were constructed by amplifying the first 240 nucleotides of the HPV16 E6 ORF (corresponding to amino acids 1 to 80) with a PCR for subsequent cloning into the relative destination vector with Gateway^®^ cloning technology (Thermo Fisher Scientific, Waltham, MA, USA) as previously described [23]. Plasmids pcDNA3-RLuc-E6 Y43E/F47R, pDEST6.2-YFP-E6 Y43E/F47R, pcDNA3-RLuc-E6_N_ Y43E/F47R, and pDEST6.2-YFP-E6_N_ Y43E/F47R were generated with the QuikChange^®^ Site-Directed mutagenesis system (Stratagene, Agilent Technologies, Santa Clara, CA, USA) as previously described [23]. Plasmids pDEST6.2-YFP-E6 T149D/L151A and pcDNA3-RLuc-E6 T149D/L151A were constructed by transferring the E6 T149D/L151A-encoding to the relative destination vector with Gateway^®^ cloning technology (Thermo Fisher Scientific, Waltham, MA, USA) as previously described [23]. Plasmid pcDNA3-RLuc-UL44 was previously reported [25]. Plasmids pK-YFP-hScrib (#58738, [26]) and pcDNA-Xpress-His-USP15 (#23216, [27]) were purchased from Addgene (Watertown, MA, USA). Plasmid pcDNA3.1-HA-HPV16E6 was previously reported [28] and plasmid pcDNA3.1-HA-HPV16E6 Y43E/F47R was generated as described above. Plasmid pcDNA3.1-HA-HPV16E6 T149D/L151A was created by transferring the HA-E6 T149D/L151A-encoding sequence into the pcDNA3.1 empty vector (Invitrogen, Thermo Fisher Scientific, Waltham, MA, USA) with HindIII and EcoRI restriction enzymes (NE BioLabs, Ipswich, MA, USA). Plasmid pDEST-FLAG-E6 was constructed by cloning the HPV16 E6 sequence into the pDEST-nFLAG vector with Gateway^®^ cloning technology (Thermo Fisher Scientific, Waltham, MA, USA) as previously described [23], while plasmid pCMV-Myc-E6 was constructed by cloning the HPV16 E6 sequence into the pCMV-Myc vector (Clontech) with EcoRI and NotI restriction enzymes (NE BioLabs, Ipswich, MA, USA). All PPI assays (BRET, CoIP, PLA) were performed with plasmids expressing splicing-defective E6 proteins, edited with the QuikChange^®^ Site-Directed mutagenesis system (Stratagene, Agilent Technologies, Santa Clara, CA, USA) as previously described [23]. Plasmid pCMV-HPV16E7-FLAG/HA was previously reported [29], while plasmids pDEST6.2-E7-YFP and pDEST6.2-Lumio-E7 were constructed by cloning the HPV16 E7 sequence into the pDEST6.2-cYFP and pDEST6.2-nLumio vectors, respectively, with Gateway^®^ cloning technology (Thermo Fisher Scientific, Waltham, MA, USA) as previously described [23].

### 2.3. Transfection Techniques and the Generation of Stable Cell Lines

HEK 293T and H1299 cells were transfected with Lipofectamine™ 2000 and Lipofectamine™ 3000 (Invitrogen, Thermo Fisher Scientific, Waltham, MA, USA), respectively, according to the manufacturer’s protocol with a constant Lipofectamine/DNA ratio of 2:1 (μL/μg). C33A cells were transfected with standard calcium-phosphate procedures with a constant CaCl_2_/DNA ratio of 30:1 (μmol/μg). HaCaT and HEKn cells were transfected with a polymeric ExGen500 transfection reagent (Biomol GmbH, Hamburg, Germany) according to the manufacturer’s protocol with a constant ExGen500/DNA ratio of 6.6:1 (μL/μg). Stable HaCaT cell lines were generated by transfecting HaCaT cells with pcDNA3.1 plasmids expressing HA-tagged E6 proteins or the empty vector. Twenty-four hours post-transfection cells were selected with 1 mg/mL geneticin (Gibco, Thermo Fisher Scientific, Waltham, MA, USA) and treated for 7 days. Transfected cells were then plated as single cells in 96-well plates and cultured with 750 μg/mL geneticin (Gibco, Thermo Fisher Scientific, Waltham, MA, USA) for monoclonal expansion. Clones were screened for transgene expression through Western blotting (see below). For the generation of HaCaT-E6+E7 cells, HaCaT-E6 cells were transfected with pDEST6.2-Lumio-E7. Twenty-four hours post-transfection cells were selected with 5 μg/mL blasticidin (Gibco, Thermo Fisher Scientific, Waltham, MA, USA) and treated for 5 days. Transfected cells were then plated as single cells in 96-well plates and cultured with 2 μg/mL blasticidin (Gibco, Thermo Fisher Scientific, Waltham, MA, USA) for monoclonal expansion. Clones were screened for transgene expression with the TC-FIAsH™ II In-Cell Tetracysteine Tag Detection Kit (Thermo Fisher Scientific, Waltham, MA, USA) according to the manufacturer’s protocol using an inverted epi-fluorescent Leica DM IL LED microscope (Leica Microsystems, Wetzlar, Germany) equipped with a digital DFC420C camera and a 10× objective.

### 2.4. Bioluminescence Resonance Energy Transfer (BRET) Data Acquisition and Analysis

BRET assays were performed as previously described [30] with minor modifications. Briefly, 10^5^ HEK 293T cells were transfected in 24-well plates with a plasmid expressing an RLuc-tagged (donor) protein alone or in combination with the plasmid expressing the relative YFP-tagged (acceptor) partner, according to the PPI examined. Given the ability of E6 to bind its own pre-mRNA to enhance its translation [31], transfections involving the expression of full-length E6 proteins were normalized with the relative p513-based E6-expressing construct in order to deliver a constant amount of E6-expressing plasmids throughout all transfections. Single-point BRET experiments were performed by transfecting the acceptor- and donor-expressing plasmids in a ratio of 100:1 (ng/ng). For BRET E6_N_ saturation experiments, fixed amounts for RLuc-tagged (donor) E6_N_-expressing plasmids were co-transfected with increasing amounts of the relative YFP-tagged (acceptor) E6_N_-expressing construct or the empty vector in the presence of 200 ng of pcDNA-Xpress-His-USP15 to increase E6_N_ protein levels as previously reported [27]. For E6 homodimerization and E6/hScrib BRET assays, data were collected when cells reached complete confluence, while for the E6/p53 BRET experiments, data were collected at subconfluence. At 24 h post-transfection, cells were treated with 40 μM of proteasome inhibitor MG-132 (Sigma-Aldrich, St. Louis, MO, USA) for 3 h to block the E6-mediated hScrib and p53 degradations and then transferred in triplicate into black Costar 96-well plates (Corning, Glendale, AZ, USA) for fluorimetric/luminometric analysis. Fluorimetric analysis was performed exposing cells at an excitation wavelength of 485 ± 15 nm, measuring YFP-tagged protein emission at 535 ± 25 nm (Em_fluo_535), integrated over 1 s, using a VICTOR X2 Multilabel Plate Reader (PerkinElmer, Waltham, MA, USA). Luminometric analysis was performed by adding benzyl-coelenterazine at a final concentration of 5 μM (P.j.k. GmbH, Kleinblittersdorf, Germany) and measuring the emission signals at 535 ± 25 nm (Em_lum_535) and 460 ± 25 nm (Em_lum_460), integrated over 1 s, at 5 and 15 min after substrate addition. BRET values (BV) were calculated for each well, after blank subtraction, as Em_lum_535/Em_lum_460. BRET ratios (mBR) were calculated for cells co-transfected with YFP- and RLuc- tagged constructs as (BV_sample_—BV_background_) × 1000 at 15 min after substrate addition, where BV_background_ represents the Em_lum_460 of cells transfected with the relative RLuc-tagged construct alone. For BRET binding curves, mBRs were plotted against Y/R values, expressed as Em_fluo_535/Em_lum_460 calculated at 5 min after substrate addition. Appropriate amounts of each RLuc-tagged construct were selected to yield a luminometric signal of about 2000 ± 500 luminescence units at 5 min after substrate addition.

### 2.5. Proximity Ligation Assays (PLA)

Five × 10^4^ H1299 cells were seeded on glass 12 mm-diameter coverslips in 24-well plates and transfected with equimolar combinations of pDEST-FLAG-E6, pCMV-Myc-E6, or empty vectors 24 h post-seeding. Forty-eight hours post-transfection, cells were fixed in 4% PFA solution (Santa Cruz Biotechnology, Dallas, TX, USA) for 10 min at room temperature, washed with PBS, and PLAs were performed with either Duolink™ In Situ Orange or Duolink™ In Situ Green detection reagents (Sigma-Aldrich, St. Louis, MO, USA) according to the manufacturer’s protocol. Images were acquired with a Nikon A1RSi Laser Scanning inverted confocal microscope equipped with NIS-Elements Advanced Research software (Nikon Instruments Inc., Tokyo, Japan) using 20× and 60× ocular objectives.

### 2.6. Immunofluorescence and Live-Imaging Analyses

Immunofluorescence studies were performed by seeding cells on glass 12 mm-diameter coverslips in 24-well plates. Cells were fixed in 4% PFA solution (Santa Cruz Biotechnology, Dallas, TX, USA) for 10 min at room temperature, washed in PBS, permeabilized with 0.3% Triton X-100 in PBS for 10 min at room temperature, washed in PBS, and blocked with 0.1% Triton X-100 in PBS (PBST) with 4% BSA for at least 1 h at room temperature. Primary antibody incubations were performed overnight at 4 °C in 2% BSA in PBST. The next day, fixed cells were washed in PBST, incubated with Alexa Fluor^®^-conjugated secondary antibodies (Thermo Fisher Scientific, Waltham, MA, USA) in 2% BSA in PBST for 1 h at room temperature, and washed with PBST. Nuclear staining was performed by incubating cells with DRAQ5™ (BioStatus Ltd., Shepshed, UK) 1:2000 in PBS for 20 min at room temperature followed by five washes in PBS. Coverslips were mounted using the FluorSave™ Reagent (Calbiochem, Sigma-Aldrich, St. Louis, MO, USA). Immunostaining of hScrib in HaCaT cells was performed using an anti-hScrib goat primary antibody (C-20, Santa Cruz Biotechnology, Dallas, TX, USA), while hScrib in CaSki and SiHa cells was stained with an anti-hScrib rabbit primary antibody (HPA064312, Sigma-Aldrich, St. Louis, MO, USA). A complete list of the antibodies used in this study can be found in Appendix A. For live imaging analyses, 2 × 10^5^ H1299 cells were seeded in 6-well plates and transfected with plasmid pDEST6.2-YFP-E6 or pDEST6.2-YFP-E6 Y43E/F47R. Cells were imaged at complete confluence using a phenol red-free DMEM medium (Thermo Fisher Scientific, Waltham, MA, USA) supplemented with 10% FBS. Nuclear (Fn) and cytoplasmic (Fc) fluorescence measurements were performed as previously described [32] but mitotic cells were excluded from the analysis. Images were acquired with a Nikon A1RSi Laser Scanning inverted confocal microscope equipped with NIS-Elements Advanced Research software (Nikon Instruments Inc., Tokyo, Japan) using 20× and 60× ocular objectives. Live imaging was performed by keeping cells at 37 °C with 5% CO_2_ in an Okolab Heating System (Okolab Inc., Ambridge, PA, USA) mounted over a Nikon A1RSi confocal microscope. Laser excitation and emission filters were: blue laser at 488 nm, emission bandwidth 500–550 nm; green laser at 561 nm, emission bandwidth 570–620 nm; and red laser at 640 nm, emission bandwidth 660–730 nm. Final images included in this work are representative of different fields of view from multiple experiments.

### 2.7. Western Blotting

Western blots were performed as previously described [23] with minor modifications. Briefly, cell pellets were lysed in a cold RIPA buffer (50 mM Tris, 150 mM NaCl, 1% IGEPAL, 0.1% SDS, pH 7.8) supplemented with a Halt Protease Inhibitor Cocktail (Thermo Fisher Scientific, Waltham, MA, USA) for 30 min in ice and soluble proteins were retrieved by centrifuging cell extracts at 16,000× *g* for 10 min at 4 °C. Insoluble proteins were extracted using a modified version of the RIPA buffer (iRIPA: 10 mM Tris, 150 mM NaCl, 1% IGEPAL, 1 mM EDTA, 0.1% SDS, 0.5% Sodium Deoxycholate, pH 7.8) for 1 h in ice and these extracts were not cleared by centrifugation. Cellular lysates were separated by SDS-PAGE, transferred onto polyvinylidene difluoride (PVDF) membranes, and analyzed by Western blotting. Untagged wild-type E6 was detected using an anti-HPV16 E6 goat primary antibody (N-17, Santa Cruz Biotechnology, Dallas, TX, USA), while for the simultaneous detection of untagged wild-type and mutant E6 proteins, we used an anti-HPV16/18 E6 mouse primary antibody (C1P5, Santa Cruz Biotechnology, Dallas, TX, USA). Tagged E6 fusion proteins were detected using anti-tag primary antibodies. Blotted proteins were visualized using Horseradish peroxidase (HRP)-conjugated secondary antibodies with an Uvitec Alliance Q9 Mini digital imager (BioSPX, Abcoude, The Netherlands). Loading controls displayed in the final images were always blotted on the same membrane where the relative protein under examination was visualized. Final images included in this work are representative of multiple experimental replicates. A complete list of the antibodies used in this study can be found in Appendix A.

### 2.8. Coimmunoprecipitation (CoIP) Assays

CoIP experiments were performed as previously described [33] with minor modifications. Briefly, Protein G PLUS-Agarose beads (Santa Cruz Biotechnology, Dallas, TX, USA) were equilibrated in a CoIP buffer (20 mM HEPES, 100 mM NaCl, 2.5 mM MgCl_2_, 5% glycerol, 1% Triton X-100, 0.5% IGEPAL, pH 7.8), incubated with an anti-hScrib primary antibody for 2 h at 20 °C on a rotating wheel and then blocked overnight in a CoIP buffer with 10% FBS on a rotating wheel at 4 °C. To immunoprecipitate hScrib from HaCaT and H1299 cell extracts we used an anti-hScrib mouse primary antibody (D-2, Santa Cruz Biotechnology, Dallas, TX, USA) and an anti-hScrib rabbit primary antibody (PA5-28628, Invitrogen, Thermo Fisher Scientific, Waltham, MA, USA), respectively. Control IPs were performed by directly blocking Protein G PLUS-Agarose beads (Santa Cruz Biotechnology, Dallas, TX, USA) in a CoIP buffer with 10% FBS without primary antibody incubation. The next day, cell pellets were lysed in a CoIP buffer supplemented with a Halt Protease Inhibitor Cocktail (Thermo Fisher Scientific, Waltham, MA, USA) and were quantified using a Pierce™ BCA Protein Assay Kit (Thermo Fisher Scientific). Cell lysates were not cleared by centrifugation throughout these experiments. Lysates were precleared using Protein A/G PLUS-Agarose beads (Santa Cruz Biotechnology, Dallas, TX, USA) for 30 min at 4 °C on a rotating wheel in a CoIP buffer prior to the immunoprecipitation with a constant lysate/beads ratio of 6.66:1 (μg/μL). Cleared lysates were added over blocked beads with a constant lysate/beads/Ig ratio of 40:8:1 (μg/μL/μg) and were incubated for 30 min on a rotating wheel at 4 °C in a CoIP buffer. Beads were retrieved, washed four times with a CoIP buffer, and precipitated proteins were eluted with 20 μL of an Elution Buffer (10 mM Tris, 300 mM NaCl, 5 mM EDTA, 0.5% SDS, pH 8) for 30 min at 65 °C. Elutions were quantified using a Pierce™ BCA Protein Assay Kit (Thermo Fisher Scientific, Waltham, MA, USA) and subjected to Western blotting.

### 2.9. Quantitative Real-Time PCR (qPCR)

qPCRs were performed as previously described [33]. Briefly, stable HaCaT cell lines were seeded on 60 mm cell culture dishes and were either untransfected or transfected with the pCMV-HPV16E7-FLAG/HA plasmid. The total RNA was purified with an RNA Purification Plus Kit (Norgen Biotek, Thorold, ON, Canada) and cDNAs were generated using random primers (Applied Biosystems) and M-MLV reverse transcriptase (Applied Biosystems, Waltham, MA, USA). The qPCR was performed with SYBR green (Applied Biosystems) on a 7900 HT Fast Real-Time PCR System (Applied Biosystems, Waltham, MA, USA). A complete list of the primer sequences used in this study can be found in Appendix A.

### 2.10. Three-Dimensional Spheroid Formation Assays

HaCaT-derived spheroids were generated by plating transfected cells as single cell suspensions in 24-well ultra-low attachment Nunclon Sphera plates (Thermo Fisher Scientific, Waltham, MA, USA). Cells were seeded 8 h post-transfection at a density of 1000 cells/cm^2^ and were grown in DMEM/F-12 (Thermo Fisher Scientific, Waltham, MA, USA) supplemented with 10 ng/mL b-FGF (Gibco, Thermo Fisher Scientific), 10 ng/mL EGF (Gibco, Thermo Fisher Scientific, Waltham, MA, USA), B27 (Gibco, Thermo Fisher Scientific, Waltham, MA, USA), 2 mM L-glutamine (Gibco, Thermo Fisher Scientific, Waltham, MA, USA), and 1 μg/mL hydrocortisone (Sigma-Aldrich, St. Louis, MO, USA). The medium was changed every 48 h and images were acquired after at least 7 days using an inverted epi-fluorescent Leica DM IL LED microscope (Leica Microsystems, Wetzlar, Germany) equipped with a digital DFC420C camera and a 10× objective.

### 2.11. Differential Gene Expression Analysis

Transcript microarray data of primary keratinocytes expressing HPV16 E6, HPV16 E7, or both [34] were downloaded from the Gene Expression Omnibus (GEO) database with accession codes GSE58841. Gene expression analyses were performed using the GEO2R online tool (https://www.ncbi.nlm.nih.gov/geo/geo2r/, accessed on 18 January 2021) and all biological replicates available within the dataset were used for the analysis. Differentially expressed genes were selected on the basis of an adjusted *p* value < 0.05.

### 2.12. Quantification and Statistical Analysis

All experiments were performed with multiple biological replicates and the statistical methods used for the analysis are mentioned in the respective Figure legends. Whenever possible, statistical analyses were performed using the mean values of independent experiments performed at least with triplicates and the relative statistical methods were performed using standard deviations (SD). When representative experiments were displayed, the statistical analysis was performed using the mean and standard error of the mean (SEM) values. Sample sizes are reported in the respective Figure legends. All statistical methods used in this study were performed without arbitrary assumptions. All statistical analyses were performed using GraphPad Prism 8. Fluorescence quantifications were performed using ImageJ 2.0.

## 3. Results

### 3.1. HPV16 E6 and E7 Are Concomitantly Required to Promote YAP/TAZ Signaling and Stemness Properties

In order to dissect the specific activities of the oncoprotein E6 that sustain the induction of YAP and TAZ, we initially decided to investigate the independent contribution of HPV E6 and E7 oncoproteins in fostering YAP/TAZ signaling. To this end, we started by monitoring the transcription of a series of bona fide YAP/TAZ-target genes in HaCaT cells overexpressing either HPV16 E6, HPV16 E7, or both (Appendix A). We chose HaCaT cells as they are non-transformed keratinocytes in which YAP/TAZ activity is physiologically regulated by upstream signals [35]. We performed the analysis through a quantitative real-time PCR (qPCR) in cells cultured at complete confluence, a condition that determines YAP/TAZ protein degradation preventing downstream YAP/TAZ-mediated gene transcription [36], and compared the behavior of E6- and/or E7-overexpressing cells versus the control HaCaT cells. Although both E6 and E7 alone could upregulate the transcription of a few YAP/TAZ-target genes (Figure 1A and Appendix A), HaCaT cells co-expressing both viral oncoproteins exhibited the highest YAP/TAZ-target gene induction (Figure 1A and Appendix A), suggesting that both E6 and E7 are required for a full biological response.

One of the major well-established consequences of full YAP/TAZ-pathway activation in non-transformed cells is the acquisition of stem-like traits [12,37]. The in vitro gold standard assay to assess the acquisition of stemness features is the generation of three-dimensional spheroids from single-cell suspensions [38]. Therefore, we tested whether HaCaT cells overexpressing E6 and/or E7 acquired stemness potential by seeding transfected cells in non-adherent cell culture conditions and monitoring the generation of spheroids. Accordingly, only cells expressing both E6 and E7 efficiently generated spheroids (Figure 1B). Considering these results, we wished to validate that only cells concomitantly expressing E6 and E7 undergo a significant reprogramming towards their lineage-specific progenitor, which has been shown to depend on YAP/TAZ functions [10,12]. We thus interrogated the Gene Expression Omnibus database and analyzed the gene expression patterns of primary keratinocytes transduced to express either HPV16 E6, E7, or both [34]. Taking advantage of the recently characterized gene signature of epithelial stem cells [39], we analyzed the transcription of epithelial stemness genes. Interestingly, we observed that only the combined expression of E6 and E7 resulted in a significant upregulation of several epithelial basal markers (Figure 1C). Taken together, these findings indicate that both E6 and E7 are simultaneously required to fully activate YAP/TAZ signaling and promote the acquisition of stemness features.

### 3.2. HPV16 E6 Forms Dimers in the Cell and Its Homodimerization Is Linked to YAP/TAZ Upregulation

It is known that E7 sustains YAP/TAZ induction through the degradation of PTPN14 [15,16], a known negative regulator of YAP [40]. To explore the mechanisms through which the oncoprotein E6 cooperates with E7 to foster YAP/TAZ signaling, we focused our attention on the activities of E6 that could directly lead to YAP/TAZ upregulation. Considering the reported p53-dependent activation of core Hippo kinases [18], and the regulatory activity of some PDZ proteins involved in epithelial polarity for YAP/TAZ functions [19,20,21], we postulated that the E6-mediated degradation of either p53 or PDZ proteins could determine YAP/TAZ upregulation. Indeed, a recent study indicated a direct role of PDZ-protein degradation induced by high-risk E6 variants in YAP nuclear retention [22], but the possible contribution of p53 degradation for YAP/TAZ induction has not been investigated yet. Therefore, we transiently overexpressed wild-type HPV16 E6, E6 Y43E/F47R (a mutant E6 protein defective for p53 degradation, [23,41]), and E6 T149D/L151A (a mutant E6 defective for PDZ-protein binding and degradation, [24]) into different HPV-negative epithelial cell lines. In order to investigate the role of p53 degradation for YAP/TAZ upregulation, we selected three cell lines based on their p53 status: human epidermal keratinoctyes (HEKn, wild-type p53), C33A cervical cancer cells (mutated p53), and H1299 lung cancer cells (p53-null). Following the transient transfection of the E6-expressing constructs, we analyzed the endogenous levels of YAP and TAZ when all cell lines reached complete confluence. Intriguingly, in HEKn cells we observed that the overexpression of wild-type E6 could sustain elevated YAP/TAZ protein levels at a high cell density (Figure 2A), while the overexpression of both E6 Y43E/F47R and E6 T149D/L151A mutants could not sustain YAP and TAZ upregulation, suggesting that the degradation of both p53 and PDZ proteins might be required to foster YAP/TAZ induction (Figure 2A). However, when we analyzed YAP and TAZ protein levels in confluent C33A and H1299 cells wherein p53 was either mutated or absent, respectively, we unexpectedly observed that the expression of E6 Y43E/F47R did not result in any upregulation of YAP/TAZ protein levels, similar to cells expressing E6 T149D/L151A (Figure 2A). Therefore, we reasoned that the degradation of p53 induced by E6 could be not involved in the mechanism of YAP and TAZ upregulation, but, in parallel to the degradation of PDZ proteins, E6 might foster YAP/TAZ through another PPI involving the same residues of E6 required for p53 binding.

Prompted by the above results, we looked for a candidate PPI involving the same N-terminal region of E6 required for the binding to p53 that corresponds to the alpha helix α2, in which residues Y43 and F47 constitute the hydrophobic binding core (Appendix A). To our knowledge, the only documented PPI involving the α2 helix of E6, other than E6/p53 binding, appears to be E6 homodimerization [42]. According to the available structural data, Y43 and F47 are two solvent-exposed residues that in vitro have been shown to participate in both p53 binding and E6 homodimerization (Figure 2B) [41,42]. Therefore, we reasoned that E6 homodimerization might represent the candidate interaction involved in YAP/TAZ upregulation, explaining why E6 Y43E/F47R cannot sustain high YAP/TAZ protein levels in p53-null cells (Figure 2A). Nevertheless, since E6 homodimerization has been previously observed only in vitro [42,43], we first wished to demonstrate that E6 can dimerize also in a cellular context, and that E6 Y43E/F47R, while being unable to bind and degrade p53, cannot also self-associate. To this aim, we took advantage of the Bioluminescence Resonance Energy Transfer (BRET) technology to study E6 homodimerization in living cells. The technique allows the detection of a PPI in cells by fusing YFP and RLuc moieties to the proteins of interest and measuring their association by means of the energy transfer from the RLuc tag (donor) to the YFP moiety (acceptor). Strikingly, through the ectopic overexpression of RLuc- and YFP-tagged E6 proteins in HEK 293T cells, we could successfully detect a reproducible E6 self-association signal, demonstrating for the first time the existence of a dimeric form of E6 in a cellular context (Figure 2C). The E6 self-interaction signal detected by the BRET assays was specific as it was significantly above the level of non-specificity measured by co-expressing YFP-tagged E6 proteins with either a RLuc-tagged unrelated viral protein or RLuc alone (Appendix A). Moreover, the homodimerization of E6 was indeed driven by its N-terminal domain (Figure 2C,D) in accordance with available structural data [42], and the substitution of Y43 and F47 with polar residues drastically impaired E6 self-association (Figure 2C,D).

In parallel, we sought to confirm the contribution of residues Y43 and F47 also for the binding of HPV16 E6 to p53 in living cells using the same experimental approach. Thus, we co-expressed RLuc-p53 and YFP-E6 in HEK293T cells for cell-based BRET measurements. Indeed, we could detect E6–p53 binding, and the interaction occurred only in cells expressing the full-length viral protein (Figure 2E, left graph), in accordance with the structural model showing the requirement of both zinc-finger domains of E6 for productive p53 binding [41]. Most importantly, the substitution of Y43 and F47 with polar residues on the surface of the E6 N-terminal domain significantly disrupted p53 binding (Figure 2E, left panel) and degradation (Figure 2E, right panel), in line with previously published in vitro data [23,41].

These data thus indicated that E6 can dimerize in cells, and E6 homodimerization is indeed driven by the same interacting surface required for p53 binding. Therefore, our results pointed to a scenario in which, in parallel to PDZ-protein degradation, E6 could sustain YAP/TAZ induction through E6 homodimerization.

### 3.3. The Dimeric Form of E6 Localizes in the Cytosol and Takes Part in the E6–hScrib Complex Formation

To try to understand how E6 homodimerization integrates into the mechanism of YAP/TAZ induction, we first investigated the localization of E6 dimers in the cell. To this aim, we performed Proximity Ligation Assays (PLA) using Myc-tagged and FLAG-tagged full-length E6 proteins overexpressed in H1299 cells (Appendix A). The technique allows the visualization of where a PPI occurs in the cell through oligonucleotide-based fluorescent PLA probes conjugated to the antibodies recognizing the proteins of interest. Taking into account our results (Figure 2C,E) and the latest E6/p53 structural model showing that E6 binds to p53 through the same interacting surface required for E6 homodimerization [41], we chose the H1299 cell line as a reporter system for these experiments since they are p53-null cells, in order to avoid any possible experimental interference of p53 binding with E6 self-association. Indeed, since the E6–p53 complex formation engages E6 in its monomeric form, it has been previously suggested that E6 homodimerization and E6–p53 interaction are two competing, mutually-exclusive events [41]. By PLA we could detect E6 dimerization which appeared to occur in the cell cytosol (Figure 3A and Appendix A). Intriguingly, we did not observe the occurrence of E6 dimers in the nucleus of p53-null H1299 cells, suggesting that even in the absence of p53, overexpressed E6 cannot dimerize in the nuclear compartment (Figure 3A). Furthermore, quantitative live imaging analyses of the localization of overexpressed YFP-tagged wild-type E6 and dimerization-defective E6 Y43E/F47R revealed that the average cytoplasmic accumulation of YFP-E6 Y43E/F47R was lower than that of YFP-E6 in the cytosol (Figure 3B). Collectively, these observations supported the notion that the dimeric form of E6 has exclusive cytoplasmic localization.

Prompted by these results, we explored the possibility that E6 dimers could be involved in the physical interaction with cytoplasmic PDZ proteins that act as direct inhibitors of YAP/TAZ. In line with this hypothesis, dimerization-defective E6 Y43E/F47R might be unable to sustain YAP/TAZ induction due to impaired PDZ-protein targeting. Since we performed the experiments using E6 of HPV16, we selected the cytoplasmic PDZ protein that is preferentially targeted by this genotypic variant in vitro, i.e., hScrib of the Scribble module, a key apico-basal polarity factor in epithelial cells [44,45]. Indeed, the Scribble polarity module exerts a strong inhibitory activity against YAP/TAZ functions in the epithelium [19,20,21]. However, it has been shown that the direct binding of E6 to hScrib involves the viral C-terminal PDZ-binding motif (PBM), which binds to the PDZ3 domain of hScrib [44], and the disruption of the PBM–PDZ interaction is sufficient to completely abolish E6–hScrib binding [46]. We thus hypothesized that although E6 binds to hScrib via a PBM–PDZ interaction, it might form a protein complex with hScrib in which E6 is present in the form of dimer. To test this hypothesis, we performed coimmunoprecipitation (CoIP) assays by pulling down endogenous hScrib in H1299 cells overexpressing HA-tagged and YFP-tagged full-length E6 proteins (Appendix A). This experimental approach allows the detection of the single E6 monomers thanks to the different molecular weight of the tags. As expected, both HA-E6 and YFP-E6 could be coimmunoprecipitated with hScrib thanks to a wild-type PBM on both proteins (Figure 3C, lane 1). Remarkably, when we co-expressed HA-E6 with the YFP-tagged PDZ-binding-defective E6 T149D/L151A mutant, YFP-E6 T149D/L151A was still retained in the HA-E6–hScrib complex (Figure 3C, lane 2). Conversely, when YFP-E6 T149D/L151A was co-expressed with the dimerization-defective HA-E6 Y43E/F47R mutant, YFP-E6 T149D/L151A was not retained in the complex (Figure 3C, lane 3). This result thus suggested that E6 binds to hScrib as a dimer.

To further confirm the evidence that the E6–hScrib interaction involves the dimeric form of E6, we performed a BRET assay between YFP-hScrib and RLuc-tagged E6 proteins. When we co-expressed YFP-hScrib with RLuc-E6 in HEK 293T cells, we could indeed detect E6–hScrib binding and the interaction signal was significantly above the level of non-specificity measured by co-expressing YFP-hScrib with a RLuc-tagged unrelated viral protein lacking a PBM (Figure 3D). As expected, no significant interaction signal was detected between YFP-hScrib and RLuc-E6 T149D/L151A (Figure 3D). However, when HA-E6 was introduced in the system, RLuc-E6 T149D/L151A could successfully interact with YFP-hScrib (Figure 3D). In contrast, we did not detect any significant association between YFP-hScrib and RLuc-E6 T149D/L151A in the presence of dimerization-defective HA-E6 Y43E/F47R (Figure 3D), confirming that E6 T149D/L151A could re-interact with hScrib through E6 dimerization. Thus, E6 self-associates in the cytoplasm and, as a dimer, forms a protein complex with hScrib, suggesting that the homodimerization of E6 could be an event intimately related to the process of hScrib targeting.

### 3.4. The Dimeric Form of E6 Is Involved in hScrib Proteasomal Degradation

To confirm that the dimeric form of E6 is directly connected to the process of hScrib degradation that promotes downstream YAP/TAZ induction, we first wished to better elucidate the fate of hScrib following its interaction with E6 in cells. It is known that E6 promotes hScrib proteasomal degradation in HPV-positive cancer cells [44,46], but this process has never been dissected in detail. Thus, we initially generated HaCaT cell lines stably overexpressing either HA-tagged wild-type HPV16 E6 or an empty vector (e.v.) as a control. When grown at 100% confluency, a situation that experimentally recapitulates the physiological hScrib localization at the basolateral junctions, we observed a significant reduction in the membrane-bound form of hScrib at the cell periphery in HaCaT-E6 cells compared to control HaCaT-e.v. cells (Figure 4A). In line with the targeting of the membrane-bound form of hScrib at the cell cortex, which orchestrates cell-to-cell junctional complexes [47], E6 dramatically reduced the insoluble protein fraction of hScrib in Western blotting experiments (Appendix A). To understand the fate of the hScrib fraction targeted by E6, we treated confluent HaCaT-E6 cells with the proteasome inhibitor MG132 for a short time (3 h) to prevent compensatory autophagic/lysosomal activation [48,49]. Intriguingly, we observed that upon proteasome inhibition hScrib reappeared in specific cytoplasmic structures and punctae resembling cytoplasmic aggresomes [50], rather than relocalizing to the cell cortex (Figure 4B). Indeed, proteasome inihibition has been shown to induce aggresome formation in several biological contexts (reviewed in [51]). Next, to exclude possible experimental artifacts due to the use of engineered HaCaT cells overexpressing E6, we also investigated the fate of hScrib after proteasome inhibition in cellular models of HPV16-driven cervical cancer endogenously expressing E6. Upon the treatment of confluent CaSki and SiHa cells with MG132, we indeed observed similar cytoplasmic hScrib aggregates recapitulating the results obtained with HaCaT-E6 cells (Figure 4C).

Finally, we wished to validate that dimeric E6 directly contributes to the process of hScrib degradation. To this end, we performed PLA/immunofluorescence experiments in HaCaT cells co-expressing Myc-E6 and FLAG-E6. Transfected cells were grown at high density and then treated with MG132. Intriguingly, we could observe the colocalization of the PLA signals, with cytoplasmic hScrib staining appearing after proteasome inhibition (Figure 4D). In addition, several PLA signals were also juxtaposed to the fraction hScrib targeted by E6 (Figure 4D), suggesting that the dimeric form of E6 indeed plays a central role in the process of hScrib proteasomal degradation.

### 3.5. HPV16 E6 Sustains YAP/TAZ Induction through Its Dimerization-Dependent Targeting of hScrib

The results shown above demonstrate that the dimeric form of E6 is directly related to the process of hScrib targeting. Therefore, our initial observation that both E6 Y43E/F47R and E6 T149D/L151A cannot sustain high YAP/TAZ protein levels (Figure 2A) is justified by the fact that E6 homodimerization and the interaction of E6 with the PDZ-protein hScrib appear to be mechanistically correlated.

It is known that the inhibitory activity of the Scribble polarity module towards YAP/TAZ functions relies on the direct interaction of the components of the Scribble module with YAP and TAZ, determining YAP/TAZ cytoplasmic sequestration and inactivation [19,20,21]. Thus, we first wished to test whether E6 Y43E/F47R, as well as E6 T149D/L151A, cannot promote downstream YAP/TAZ upregulation due to the retention of YAP and TAZ in the polarity complex. To this end, we generated HaCaT cell lines stably overexpressing either HPV16 E6 Y43E/F47R or E6 T149D/L151A (Appendix A). Therefore, we performed CoIP experiments in confluent HaCaT-e.v., HaCaT-E6, HaCaT-E6 Y43E/F47R, and HaCaT-E6 T149D/L151A cells by immunoprecipitating endogenous hScrib and analyzing whether E6, YAP, and TAZ were still retained in the complex. In control HaCaT-e.v. cells, we recapitulated previous evidence showing that in contact-inhibited cells the Scribble module sequesters YAP and TAZ to promote their inactivation (Figure 5A, lane 1) [19,21]. In HaCaT-E6 cells, as expected, we observed that E6 interacted with hScrib and hence YAP and TAZ were released from polarity-inhibition (Figure 5A, lane 2). Remarkably, in HaCaT-E6 Y43E/F47R cells, although dimerization-defective E6 Y43E/F47R could still interact with hScrib thanks to the presence of the PBM, YAP, and TAZ were retained in the protein complex (Figure 5A, lane 3). Similarly, in HaCaT-E6 T149D/L151A cells, since E6 T149D/L151A could not bind to hScrib, albeit being still able to dimerize, the hScrib–YAP/TAZ complex remained unaffected (Figure 5A, lane 4). This result thus indicates that both E6 homodimerization and E6–hScrib interaction are required to release YAP/TAZ from the inhibition induced by the Scribble module. We thus propose a new mechanistic model according to which the E6-mediated perturbation of the polarity-determinant hScrib requires dimeric E6 to properly release YAP and TAZ (Figure 5B). To confirm this, we performed qPCR experiments in confluent stable HaCaT cell lines and analyzed the transcription of YAP/TAZ-target genes. Since we observed that E7 is required in conjunction with E6 to orchestrate a complete YAP/TAZ-related response (Figure 1A), we transiently overexpressed HPV16 E7 in E6-expressing stable HaCaT cell lines (Appendix A). Then, we compared the pattern of YAP/TAZ-target gene transcription in E7-expressing HaCaT-E6, HaCaT-E6 Y43E/F47R, and HaCaT-E6 T149D/L151A versus control HaCaT-e.v. cells. Indeed, upon E7 overexpression, in HaCaT-E6 Y43E/F47R and HaCaT-E6 T149D/L151A cells, several YAP/TAZ-target genes were not significantly upregulated compared to cells co-expressing wild-type E6 and E7 (Figure 5C and Appendix A). We then tried to generate three-dimensional spheroids from stable E6-expressing HaCaT cell lines also co-expressing E7. To efficiently monitor E7 overexpression during the course of these experiments, we transiently overexpressed a YFP-tagged form of E7. In line with the qPCR results, upon E7-YFP overexpression, only HaCaT-E6 cells generated spheroids, while HaCaT-E6 Y43E/F47R, and HaCaT-E6 T149D/L151A cells failed to efficiently proliferate and form spheroids in suspension (Figure 5D).

Finally, to further validate our findings, we engineered HaCaT-E6 cells to stably overexpress Lumio-tagged wild-type HPV16 E7 (termed hereafter as HaCaT-E6+E7) (Appendix A). Remarkably, the overexpression of YFP-hScrib in HaCaT-E6+E7 cells reversed their capacity to form spheres almost completely (Figure 5E and Appendix A), recapitulating the importance of hScrib degradation induced by dimeric E6 to release YAP/TAZ and foster stem-like traits.

Collectively, our results indicate that the activity of E6 required to promote YAP/TAZ signaling in concert with E7 is the dimerization-dependent targeting of the polarity factor hScrib.

## 4. Discussion

In this study, we tried to dissect the activities of the HPV oncoprotein E6 that directly sustain YAP/TAZ induction in concert with E7 in cancer cells. Our work was inspired by the observation that the precise mechanism through which E6 fosters YAP/TAZ signaling remained to be fully elucidated. All E6 functions known to date are exclusively related to the direct interaction of E6 with cellular factors, through which E6 can hijack many host biological processes in order to support the viral life cycle, but which also promote cellular transformation [52]. In the context of YAP/TAZ upregulation induced by E6 in cancer, it has been previously suggested that E6 could induce YAP through the downregulation of SOCS6, a known negative regulator of YAP [53]. However, it was later demonstrated that SOCS6 is not an interacting partner of E6 [9]. Therefore, the observed E6-mediated downregulation of SOCS6 could be a downstream consequence of the activity of E6 in promoting YAP/TAZ signaling rather than an upstream event required to upregulate YAP and TAZ. We thus postulated that other PPIs of E6 with cellular proteins could be responsible for the induction of YAP/TAZ levels and activity. Given the high number of E6-interacting partners described to date [54], we focused our attention on two mechanisms that directly converge on Hippo signaling to physiologically regulate YAP/TAZ in the absence of E6, i.e., the p53 pathway and cell polarity [18,55], which are profoundly altered in HPV-transformed cells. It is known that E6 impairs p53 signaling and epithelial polarity through the direct interaction with p53 and PDZ proteins, respectively, leading to their proteasomal degradation [52]. However, several PDZ proteins involved in cell polarity were shown to be targeted by E6 with differential genotype-specific affinities [45], although this phenomenon was extensively studied in vitro and uncertainty remains about its real significance in vivo [56]. Nevertheless, we selected the PDZ-protein hScrib of the Scribble polarity module as it is the preferential PDZ target of HPV16 E6 in vitro, the high-risk variant we used throughout this study [44,45]. Here, we showed that: (i) overexpression of HPV16 E6 invariably determines E6–hScrib association in epithelial cells (Figure 2C,D and Figure 5A) leading to hScrib proteasomal degradation (Figure 4A,B), and (ii) the E6-mediated targeting of hScrib fosters downstream YAP/TAZ signaling independently of p53 degradation.

In addition, we serendipitously discovered that the E6-mediated targeting of the PDZ-protein hScrib involves the dimeric form of E6, whose functional role has remained totally elusive so far. Indeed, it has been previously proposed that dimeric E6 could be required for efficient p53 degradation [42]. However, a later structural model indicated that E6 is actually bound to p53 as a monomer [41], thus leaving the functional role of dimeric E6 an open question. Here we present evidence that the homodimerization of E6 is an event mechanistically related to the proteasomal degradation of hScrib, and the targeting of hScrib by dimeric E6 is responsible for promoting downstream YAP/TAZ upregulation. Indeed, we show that in the absence of E6 homodimerization, YAP and TAZ are not released from the inhibition induced by the Scribble module (Figure 5A). Nevertheless, future investigations are required to elucidate the precise molecular events in which dimeric E6 exerts its function for targeting hScrib to the proteasomes and the contribution of this phenomenon to cancer progression in vivo. In this regard, we speculate that the E6-mediated dimerization-dependent targeting of hScrib may be relevant during the progression from low-grade neoplasias to high-grade malignant lesions rather than during a productive infection. Accordingly, a previous interactomic study failed to consistently detect E6–hScrib binding in retrovirally-transduced keratinocytes expressing high-risk E6 proteins [57], and hScrib levels were found to be unaltered in histological samples of warts and benign lesions [58]. However, hScrib protein levels were dramatically low in high-grade and invasive cervical cancer samples [58]. This speculation is also indirectly supported by previous evidence that showed the importance of hScrib loss for E6/E7-mediated tumor formation in xenograft mouse models [59].

An important consequence of this work is the demonstration that, in the cell, the interacting surface of E6 corresponding to the α2 helix can serve as a promiscuous platform driving two distinct PPIs, and that E6 can behave either as a monomer or a dimer in order to fulfill different tasks. Indeed, the capacity of E6 to interact with a multitude of cellular targets, while being small in size and relatively simple in structure, suggests an intrinsic plasticity essential to accomplish a wealth of activities. This work reinforces this notion and extends our understanding about the mode of action of E6 acting as a promiscuous binder, showing that E6 can also exploit a structural rearrangement into a dimeric form in order to maximize its functionality.

In line with these observations, previous studies have already demonstrated that other surface-exposed regions of E6 can drive different PPIs, all sustained by the critical conservation of specific residues that confer the ability to interact with a defined set of host proteins (reviewed in [60]). Intriguingly, the residues that constitute the hydrophobic core of the interacting surface mediating both E6 homodimerization and E6–p53 binding, particularly Y43 and F47 that crucially contribute to both interactions (Figure 6A), showed the least variability in a panel of HPV16-driven malignant lesions [61], and a high structural constraint among canonical E6 sequences of different high-risk genotypes (Figure 6B).

In conclusion, our data reveal the crucial role of dimeric E6 in hScrib targeting and degradation for downstream YAP/TAZ induction. Thus, this work highlights the importance of the N-terminal interacting surface of E6 that is required not only for p53 binding but also for E6 homodimerization, thus opening new perspectives for the development of E6 inhibitors that act by binding this highly conserved interacting surface and blocking both interactions [62,63].

## Figures and Tables

**Figure 1 cancers-13-04083-f001:**
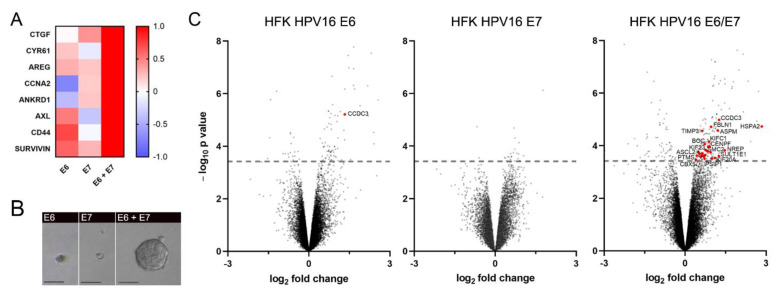
HPV16 E6 and E7 are concomitantly required to promote YAP/TAZ signaling and stemness properties. (**A**) YAP/TAZ-target gene expression profiles from qPCR data of confluent HaCaT cells overexpressing either HPV16 E6, E7, or both. The heatmap shows the normalized mean fold-change values from three independent experiments of cells overexpressing E6 and/or E7-FLAG/HA compared to confluent parental HaCaT cells. See Appendix A for non-normalized fold-change values. (**B**) Representative images of E6- and/or E7-expressing HaCaT cells grown in non-adherent cell culture conditions for the generation of spheroids. Scale bars: 50 μm. (**C**) Volcano plots of gene-expression changes in primary keratinocytes expressing either HPV16 E6, HPV16 E7, or both versus cells (accession code: GSE58841, [34]). Red dots represent significantly upregulated epithelial basal stem cell markers. The grey dashed line represents the threshold of significance (adjusted *p* value < 0.05). HFK: human foreskin keratinocytes.

**Figure 2 cancers-13-04083-f002:**
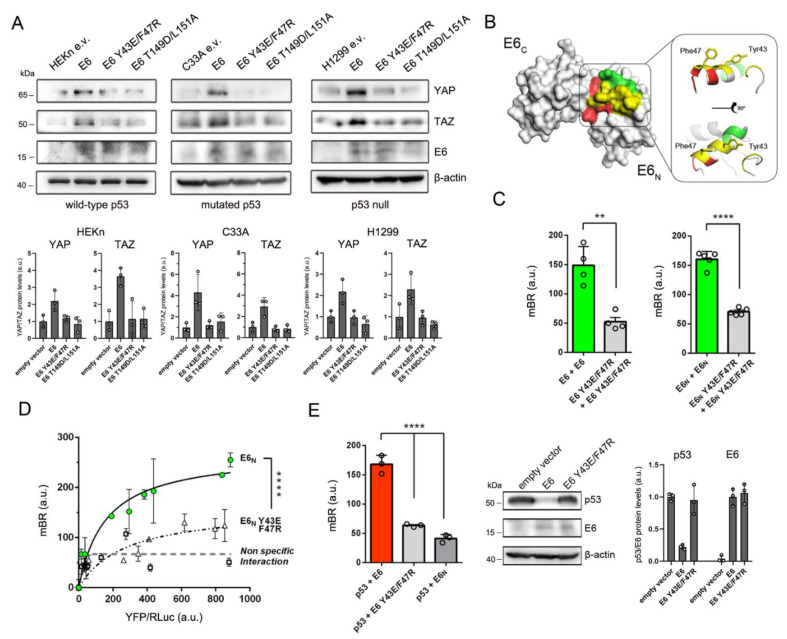
HPV16 E6 forms dimers in the cell and its homodimerization is linked to YAP/TAZ upregulation. (**A**) Western blot analysis of three different cell lines cultivated at complete confluence (HEKn: neonatal human epidermal keratinocytes, bearing wild-type p53; C33A: HPV-negative cervical cancer cells, mutated p53; H1299: HPV-negative lung cancer cells, p53-null) showing endogenous YAP and TAZ accumulation only in the presence of wild-type E6 expression, independently of cellular p53-status. β-actin was used as a loading control. Bar graphs (bottom) show the relative densitometry quantifications of YAP/TAZ protein bands normalized to β-actin. Data are Mean ± SD of 3 independent experiments. The uncropped Western blots have been shown in Appendix A. (**B**) Surface representation of E6 with residues of the α2 helix involved in E6 homodimerization (green), E6–p53 interaction (red) or both (yellow) highlighted according to published structural data [41,42]. Right panel shows the ribbon representation of the α2 helix and solvent-exposed Tyr43 and Phe47. E6_N_ and E6_C_: N- and C-terminal domains. Model created with PyMOL (PDB code: 4xr8). (**C**) E6 homodimerization measured through BRET assays expressing RLuc- and YFP-tagged full-length E6 proteins (left graph) or E6_N_ recombinant fragments (right graph) in HEK 293T cells. Data are Mean ± SD of ≥ 4 independent experiments performed in triplicate. ** *p* < 0.01, **** *p* < 0.0001 determined with unpaired two-tailed *t* tests. (**D**) BRET binding curves of the self-associations of RLuc- and YFP-tagged E6_N_ (green circles) and E6_N_ Y43E/F47R (white triangles) protein fragments expressed in HEK 293T cells. Black and dashed lines represent the non-linear regression binding curves of E6_N_ (*R^2^* = 0.937) and E6_N_ Y43E/F47R (*R^2^* = 0.836), respectively. The level of non-specific interaction (grey dashed line) represents the interaction signals (white squares) of E6_N_ with an unrelated protein (UL44 of human cytomegalovirus). Data are Mean ± SEM of a representative experiment performed with triplicates. **** *p* < 0.0001 determined with extra sum-of-squares F test. (**E**) Left graph: E6–p53 interaction measured through BRET assays in HEK 293T cells. The E6_N_–p53 interaction was used as a negative control. Data are Mean ± SD of three independent experiments performed in triplicate. **** *p* < 0.0001 determined with one-way ANOVA with Dunnett’s multiple-comparisons test. Right panel: Western blot analysis of HEK 293T cells showing endogenous p53 levels following E6 or E6 Y43E/F47R overexpression. β-actin was used as a loading control. Bar graph shows relative densitometry quantifications of p53 and E6 protein bands normalized to β-actin. Data are Mean ± SD of 3 independent experiments. In (**A**,**C**–**E**), a.u.: arbitrary units.

**Figure 3 cancers-13-04083-f003:**
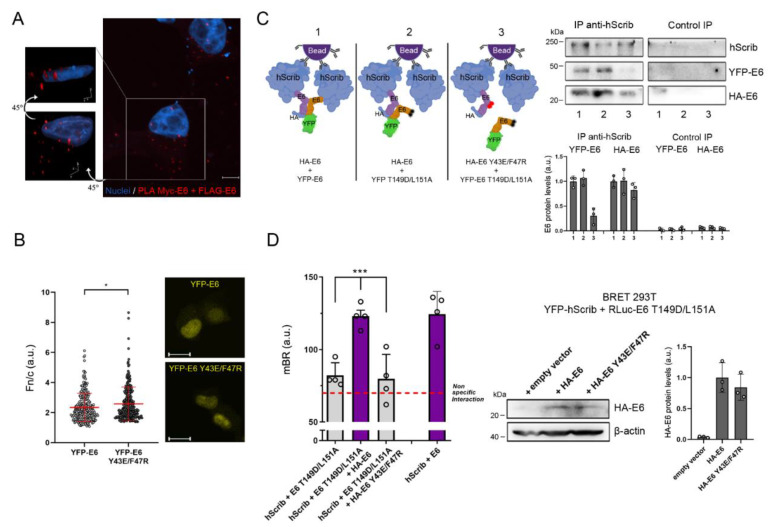
The dimeric form of E6 localizes in the cytosol and takes part in the E6–hScrib complex formation. (**A**) Representative confocal image of the Myc-FLAG Proximity Ligation Assay (PLA) detecting E6 homodimerization (red fluorescence) in H1299 cells co-expressing Myc-E6 and FLAG-E6. The image represents the maximum intensity projection of a Z-stack acquired with a 2000× magnification. Scale bar: 5 μm. Left panels show the three-dimensional reconstruction of the indicated cell and both images show the same cell, with a 45 degrees rotation around the horizontal axis between the two images. Nuclei were stained with DRAQ5. Experimental controls are shown in Appendix A. (**B**) Quantitative nuclear/cytoplasmic fluorescence (Fn/c) analysis of H1299 living cells overexpressing YFP-E6 or YFP-E6 Y43E/F47R. Data are presented as scatter-dot-plots of n_YFP-E6_ = 247 cells and n_YFP-E6 Y43E/F47R_ = 251 cells. * *p* < 0.05 determined with the Mann–Whitney U test. Right panels show the representative images used for quantification. Scale bars: 20 μm. (**C**) CoIP/Western blot analysis (right) of H1299 cells overexpressing YFP-tagged and HA-tagged E6 proteins showing the binding of tagged E6 to hScrib through immunoprecipitation of endogenous hScrib. Control IP represents the extent of non-specific binding of each protein in the experiment. Left diagram provides a schematic representation of the result obtained with the experiment, with red and black circles on E6 models indicating amino acid substitutions in the dimerization interface and PBM, respectively. Bar graphs (bottom) show relative densitometry quantifications of YFP-E6 and HA-E6 protein bands normalized to the respective precipitated hScrib. Data are Mean ± SD of 3 independent experiments. See Appendix A for whole-cell extracts input controls. (**D**) Measurement of E6 T149D/L151A interaction with hScrib through BRET assays in HEK 293T cells in the absence or the presence of HA-E6 or HA-E6 Y43E/F47R (lower graph). The dashed line indicates the level of non-specific interaction of hScrib with an unrelated protein (UL44 of human cytomegalovirus). The interaction between hScrib and wild-type E6 was used as a positive control. Data are Mean ± SD of four independent experiments performed in triplicate. *** *p* < 0.001 determined with one-way ANOVA with Dunnett’s multiple-comparisons test. Right panel shows the Western blot analysis of HEK 293T cells subjected to BRET measurements, demonstrating the presence of the corresponding HA-tagged E6 protein in transfected cells. β-actin was used as a loading control. The bar graph shows the relative densitometry quantifications of HA-E6 protein bands normalized to β-actin. Data are Mean ± SD of 3 independent experiments. In (**B**–**D**), a.u.: arbitrary units.

**Figure 4 cancers-13-04083-f004:**
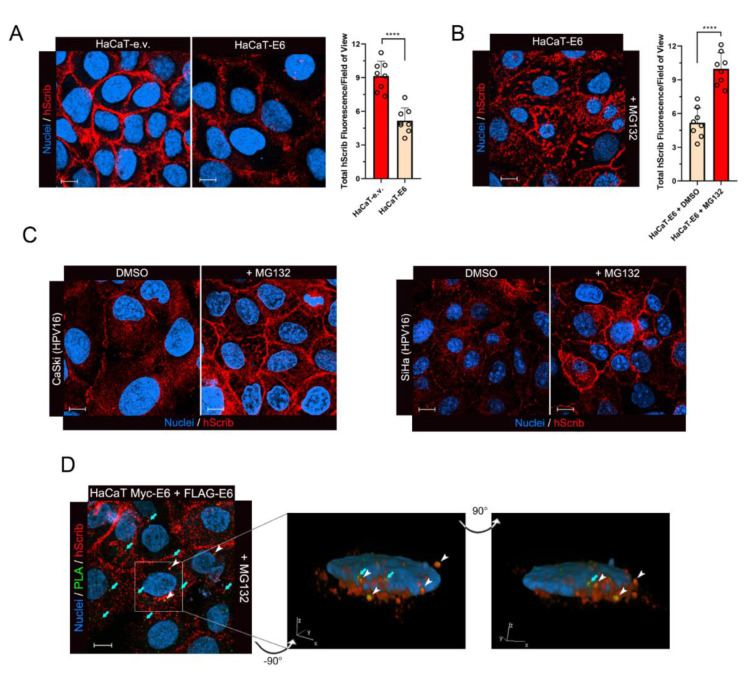
The dimeric form of E6 is involved in hScrib proteasomal degradation. (**A**) Representative immunofluorescence images of confluent HaCaT-e.v. and HaCaT-E6 cells showing endogenous hScrib localization. The right graph shows the quantification of the total hScrib fluorescence of cells cultivated at complete confluence from eight different fields of view for each sample with n_HaCaT-e.v._ = 208 cells and n_HaCaT-E6_ = 204 cells. Integrated fluorescence values were normalized over the size of selected areas used for quantification. **** *p* < 0.0001 determined with an unpaired two-tailed *t* test. (**B**) Representative immunofluorescence image of confluent HaCaT-E6 cells showing cytoplasmic hScrib rescue after treatment with proteasome inhibitor MG132 (40 μM for 3 h). The right graph shows the quantification of the total hScrib fluorescence of DMSO-treated and MG132-treated HaCaT-E6 cells cultivated at complete confluence from eight different fields of view for each sample with n_HaCaT-E6 + DMSO_ = 378 cells and n_HaCaT-E6 + MG132_ = 362 cells. Integrated fluorescence values were normalized over the size of selected areas used for quantification. **** *p* < 0.0001 determined with unpaired two-tailed *t* test. (**C**) Representative immunofluorescence images of confluent HPV16-positive CaSki and SiHa cells showing cytoplasmic hScrib rescue after treatment with the proteasome inhibitor MG132 (40 μM for 3 h) compared to vehicle-treated controls (DMSO for 3 h). (**D**) Representative Myc-FLAG PLA/immunofluorescence image of MG132-treated confluent HaCaT cells co-expressing Myc-E6 and FLAG-E6, showing both colocalization (green and red fluorescence present in the same pixels, white arrowheads) and juxtaposition (green pixels in close proximity to red pixels, cyan arrows) of E6 dimers (green fluorescence) with cytoplasmic hScrib (red fluorescence) rescued after proteasome inhibition. The right panels show the three-dimensional reconstruction of the indicated cell and both images show the same cell, with a 90 degrees rotation around the vertical axis between the two images. In all panels nuclei were stained with DRAQ5, scale bars: 10 μm.

**Figure 5 cancers-13-04083-f005:**
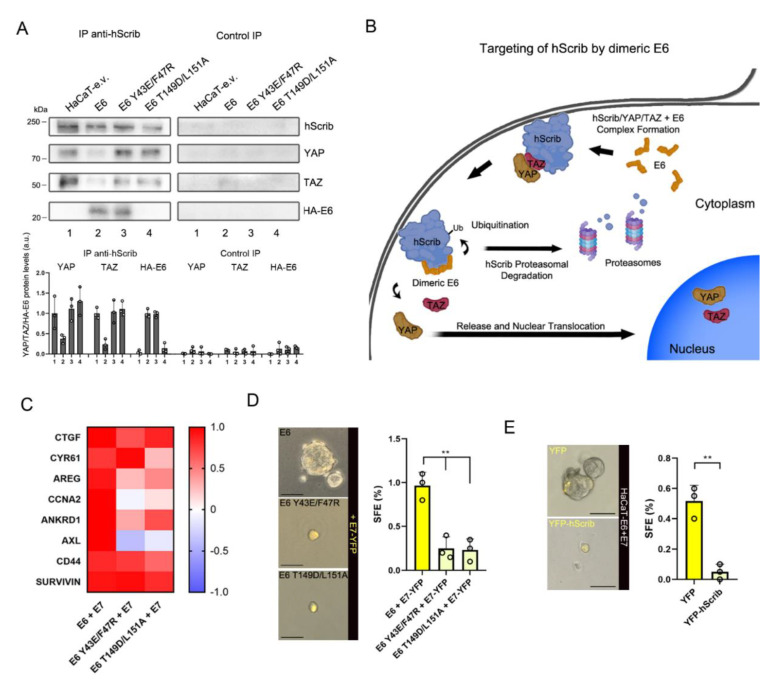
HPV16 E6 sustains YAP/TAZ induction through its dimerization-dependent targeting of hScrib. (**A**) CoIP/Western blot analysis of stable HaCaT cell lines cultivated at complete confluence showing the formation of the E6–hScrib protein complex with or without YAP/TAZ depending on the relative HA-E6 protein expressed. Control IP represents the extent of non-specific binding of each protein in the experiment. The bar graphs (bottom) show the relative densitometry quantifications of YAP, TAZ and HA-E6 protein bands normalized to the respective precipitated hScrib. Data are Mean ± SD of 3 independent experiments. See Appendix A for whole-cell extracts input controls. a.u.: arbitrary units. (**B**) Model of the proposed mechanism of hScrib targeting and consequent YAP/TAZ induction mediated by dimeric E6. Ub: Ubiquitin. (**C**) YAP/TAZ-target gene expression profiles from qPCR data of HaCaT-E6, HaCaT-E6 Y43E/F47R, and HaCaT-E6 T149D/L151A cells co-expressing HPV16 E7-FLAG/HA cultivated at complete confluence. The heatmap shows the normalized mean fold-change values from three independent experiments compared to confluent HaCaT-e.v. control cells. See Appendix A for non-normalized fold-change values. (**D**) Representative images of HaCaT-E6, HaCaT-E6 Y43E/F47R, and HaCaT-E6 T149D/L151A cells co-expressing HPV16 E7-YFP grown in non-adherent cell culture conditions for the generation of spheroids. Scale bars: 50 μm. Right graph shows the quantification of the spheres formed by the three cell lines. Only yellow-fluorescent spheroids were counted and objects with a diameter <30 μm were excluded from the analysis. ** *p* < 0.01 determined with one-way ANOVA with Dunnett’s multiple-comparisons test. (**E**) Representative images of HaCaT-E6+E7-derived spheroids overexpressing exogenous YFP or YFP-hScrib. Scale bars: 50 μm. Right graph shows the quantification of spheroids formed by HaCaT-E6+E7 transiently overexpressing YFP or YFP-hScrib. Only yellow-fluorescent spheroids were counted and objects with a diameter <30 μm were excluded from the analysis. ** *p* < 0.01 determined with unpaired two-tailed *t* test. In (**D**,**E**) SFE: Sphere-Forming Efficiency.

**Figure 6 cancers-13-04083-f006:**
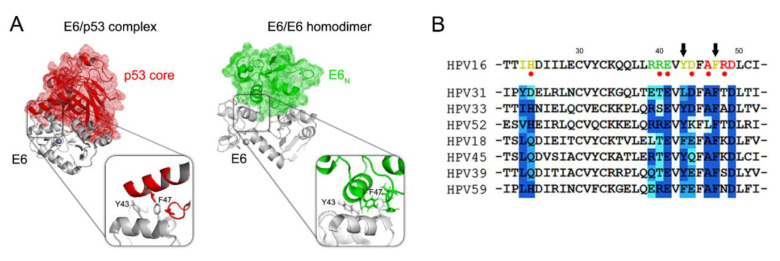
Highly conserved hydrophobic residues of the α2 helix of HPV16 E6 mediate E6–p53 and E6–E6 interactions. (**A**) Ribbon/surface representations of the E6–p53 protein complex (left model; PDB code: 4xr8) and E6 homodimer (right model; PDB code: 2ljy). Panels show the contribution of Y43 and F47 residues for both interactions. In the model of E6 homodimer, the crystal structure of full-length HPV16 E6 (PDB code: 4giz) was superimposed on one of the two E6_N_ monomers. Models created with PyMOL. (**B**) Alignment of the N-terminal amino acid residues (from 21 to 52 relative to HPV16 E6) of canonical high-risk E6 sequences. Residues of the α2 helix involved in E6 homodimerization (green), E6–p53 interaction (red) or both (yellow) are highlighted. Red circles indicate the α2 helix residues varying between HPV16 E6 variants in cervical cancer samples [57]. Black arrows indicate the critically conserved Y43 and F47 amino acids. Sequence alignment is shown according to Grantham’s distance coefficients, from dark blue (same or similar residues) to light blue (structurally dissimilar). Alignment generated with ClustalX.

## Data Availability

This study did not generate datasets. The microarray data of primary keratinocytes from Gyöngyösi et al. were downloaded from GEO under accession code GSE58841.

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
