# Peer review of "The Dimeric Form of HPV16 E6 Is Crucial to Drive YAP/TAZ Upregulation through the Targeting of hScrib"

_cancers, 2021, doi:10.3390/cancers13164083_

Round 1

Reviewer 1 Report

This is an excellent study describing the role for HPV E6 to drive YAP/TAZ upregulation by targeting hScrib for degradation. They show that the targeting of hScrib by E6 requires the homodimerization of E6, which is the first time that the dimerization of E6 has a function in cells.

Author Response

Reviewer #1:

This is an excellent study describing the role for HPV E6 to drive YAP/TAZ upregulation by targeting hScrib for degradation. They show that the targeting of hScrib by E6 requires the homodimerization of E6, which is the first time that the dimerization of E6 has a function in cells.

We are grateful to the Reviewer for her/his very positive comment and for defining our study excellent. This Reviewer did not see that anything additional was needed for acceptance, thus she/he did not request any change of the manuscript or further experiment.

Reviewer 2 Report

  This is an interesting study and an easy to read manuscript.  The quality of the data and figures is good.  In a series of experiments, the authors demonstrate that overexpression of E6 and E6 mutants supports the observation of E6 multimerization through a surface patch of E6 that also is required for interaction of E6 with p53.  This multimerization domain was initially demonstrated some years ago using bacterially expressed and purified E6.  Mutation of this surface patch was required to produce concentrated preparations of E6 necessary for crystallization of the protein.  However, showing that multimerization occurs in vivo under physiologic expression has not been clearly demonstrated as yet.   The authors overexpress E6 in various cell lines and obtain data consistent with their interpretation, that the E6 N-terminal domain self-associates, and then claim that this self association occurs only in the cytoplasm, and facilitates the the interaction of E6 with scribble and targets the degradation of scribble to activate YAP/TAZ.  The experiments overall support the claims.      But the problem is that all of the data arises from overexpression.  In vitro, E6 associates with scrib and targets scrib for degradation in rabbit reticulocyte lysates.  But the data supporting the degradation of scribble by E6 at physiologic expression levels is tenuous.  In primary keratinocytes expressing E6 from episomal genomes or from retroviruses, scrib levels are not convincingly reduced.  In TAP experiments performed by White/Howley, it was the absence of PDZ associated proteins that was more striking than the presence.  Although scrib is typically thought of as a tumor suppressor (because it really is so in drosophila), scrib expression is not consistently reduced or mutated in human cancers, and in fact, it tends to be overexpressed in human cancers.  The entire field of papilloma-virologists has been willfully trying to force E6 to target scrib for degradation in vivo for decades despite the absence of compelling evidence that this actually happens during infections, and this study does not alter that.  It is further complicated by the fact that the dimerization interface overlaps the p53 interaction domain, and while E6 reduces the concentration of p53 in HPV infected cells, it does not eliminate p53 or the association of p53 with E6----(this is observed in the White/Howley work that demonstrated abundant p53 association with E6 under conditions where very few PDZ protein peptides were recovered). So it is likely that persistent undegraded p53 could compete any E6 self-association unless performed under conditions were no p53 is present---which is what the authors did using 1299 cells.  This makes the conclusions uncertain.   So what to do with this paper?  The authors have done a lot of work here and much of it is clever.  But they evade and ignore these issues of overexpression and make unwarranted assumptions about the role of E6 interaction with scrib.   I think the work is publishable but only if the authors forthrightly confront these issues and couch the abstract, the entire discussion with the uncertainties that remain.  They are really sticking their necks out when it is unclear what the role of E6-scrib association is at all.  For example, loss of scrib has been shown to sensitize cells to apoptotic cell death through cell competition.  Why would E6 want to do that?  That makes no sense and is the opposite of what the authors are apparently thinking.         Fig. 1.  In this experiment, HaCat cells that over-express E6 and E7 from the CMV promoter are assessed for downstream effects upon YAP/TAZ responsive genes.  This demonstrates that overexpresssion can induce downstream effects, it does not show that they do this at E6 and E7 expression levels encountered in either episomal primary or cervical cancer cell lines.  This is a general flaw of overexpression in the study and a lack of comparison to either natural infection or cervical cancers.  In part B, there is no comparison to vector transfected HaCat cells which is a flaw as we do not know if either E6 or E7 alone induce anchorage independence ---- the assay does demonstrate dramatic synergy of E6 and E7 expressed together.  Similarly, in part C there is neither a negative vector control nor a positive control with an activated YAP/TAZ to compare to E6 and E7. The paper leads us to the conclusion that the induced genes are due to YAP/TAZ activities, but this is unproven.     Fig. 2.  In part A, it is notable that C33A cells, although p53 mutated, have a p53 mutation that is degraded  by 16E6, unlike most p53 mutants  (published by Peter Howley's lab).  It is notable that both pdz and F47R mutations (defective for degrading p53) both ablate the stabilization of YAP at confluency.  Once again, these are overexpression experiments that show that E6 can do this, and not that it does do this under more physiologic conditions.  Just how much E6 is being made in these cells?  A comparison to E6 expression levels from primary keratinocytes harboring HPV16 genomes should be shown.  The entire manuscript is directed at the proposal   In part C, overexpression of E6 proteins to observe E6 dimerization through amino-terminal  E6 domaine interaction.  Transient overexpression of  Luc fused E6 together with YFP fused E6 results in a signal that is ablated by mutation of F47.  The dimerization of E6 through amino terminal interaction had been previously demonstrated in vitro, and results in the in vitro aggregation of E6 preparations upon concentration of the samples.  This is the first evidence for dimerization in vivo that I know of, but there is a problem in that F47 is required for the association of E6 with p53, and thus E6 could be multimerizing within HEK 293 cells vial association with p53 tetramers.  If the assay was performed in p53 null cells there would be more confidence. In part C E6N associates with itself and that is lost upon mutation of F47 while E6N and p53 association are not observed in part E.  Again, such overexpression assays show that this self-association can occur in overexpression but does not show that it happens in physiologic conditions.    Indeed on lines 22-25 in results the authors state "These data thus indicated that E6 exists as a dimer in cells, and E6 homodimerization 22 is indeed driven by the same interacting surface required for p53 binding. Therefore, our 23 results pointed to a scenario in which, in parallel to PDZ-protein degradation, E6 could 24 sustain YAP/TAZ induction through E6 homodimerization."  Actually that statement would be more accurate if they said "E6 can exist as a dimer in overexpression conditions using a surface that overlaps with the surface required for p53 interaction."  At this point thee is no data implicating this in YAP/TAZ overexpression.     Fig 3.  The PLA data in part A lacks positive and especially negative controls.  In part B the data may reach significance but it is not exactly compelling.  There are not other mutants included that all show no difference and the differences that are shown are small.  It may be significant, but is it important?   

Author Response

Reviewer #2:

This is an interesting study and an easy to read manuscript.  The quality of the data and figures is good.  In a series of experiments, the authors demonstrate that overexpression of E6 and E6 mutants supports the observation of E6 multimerization through a surface patch of E6 that also is required for interaction of E6 with p53.  This multimerization domain was initially demonstrated some years ago using bacterially expressed and purified E6.  Mutation of this surface patch was required to produce concentrated preparations of E6 necessary for crystallization of the protein.  However, showing that multimerization occurs in vivo under physiologic expression has not been clearly demonstrated as yet.   The authors overexpress E6 in various cell lines and obtain data consistent with their interpretation, that the E6 N-terminal domain self-associates, and then claim that this self association occurs only in the cytoplasm, and facilitates the the interaction of E6 with scribble and targets the degradation of scribble to activate YAP/TAZ.  The experiments overall support the claims.

We thank the Reviewer for judging our study interesting and for commenting positively that “the quality of the data and figures is good” and “the experiments overall support the claims”.

But the problem is that all of the data arises from overexpression.  In vitro, E6 associates with scrib and targets scrib for degradation in rabbit reticulocyte lysates.  But the data supporting the degradation of scribble by E6 at physiologic expression levels is tenuous.  In primary keratinocytes expressing E6 from episomal genomes or from retroviruses, scrib levels are not convincingly reduced.  In TAP experiments performed by White/Howley, it was the absence of PDZ associated proteins that was more striking than the presence. 

We agree with the point raised by the Reviewer regarding previous studies reported in the literature by different groups, which are controversial and could derive from different experimental conditions. We emphasized in the manuscript that hScrib was selected being the preferential target of HPV16 E6 in vitro (page 11, line 29; page 17, line 25; page and line numbers refer to the cleaned manuscript with hidden track changes). We replied to the Reviewer’s concern about the use of overexpression system in the following responses.

Although scrib is typically thought of as a tumor suppressor (because it really is so in drosophila), scrib expression is not consistently reduced or mutated in human cancers, and in fact, it tends to be overexpressed in human cancers.

We slightly disagree with this opinion as several papers published in top journals have invariably demonstrated that Scrib acts indeed as a tumor suppressor not only in Drosophila but also in mammalian cells (Humbert et al., Oncogene 2008; Cordenonsi et al., Cell 2011; Yang et al., PNAS 2015; Mohseni et al., Nat Cell Biol 2014). In addition, reports of its tumor suppressive functions also in advanced cancers are emerging (Kapil et al., Oncotarget 2017). Contextualized to our work, we think that the targeting of hScrib mediated by dimeric E6 could have significance during the process of cellular transformation from low-grade to high-grade tumors as hScrib was shown to diminish in histopathological cervical cancer samples (Nakagawa et al., Br J Cancer 2004) and to be reduced also in cervical cancer-derived cell lines (Simonson et al., Cancer Res 2005).

The entire field of papilloma-virologists has been willfully trying to force E6 to target scrib for degradation in vivo for decades despite the absence of compelling evidence that this actually happens during infections, and this study does not alter that.  It is further complicated by the fact that the dimerization interface overlaps the p53 interaction domain, and while E6 reduces the concentration of p53 in HPV infected cells, it does not eliminate p53 or the association of p53 with E6----(this is observed in the White/Howley work that demonstrated abundant p53 association with E6 under conditions where very few PDZ protein peptides were recovered). So it is likely that persistent undegraded p53 could compete any E6 self-association unless performed under conditions were no p53 is present---which is what the authors did using 1299 cells.  This makes the conclusions uncertain.  

We agree with the objection that several PDZ targets are likely not really bound by E6 in physiological situations, such as during a productive infection, and we are aware that some interactomic studies failed to observe the association of E6 with Scrib or Dlg (White et al., J Virol 2012; Rozenblatt-Rosen et al., Nature 2012). However, in our experience one thing is infection, another thing is transformation. Although both are characterized by biological phenomena sustained by E6 and E7, there are subtle differences, and their deregulated expression in cancer suggests to take into consideration protein-protein interactions that may have rather minor/no significance in the context of a productive viral life-cycle, such as the interaction with hScrib. Moreover, it was previously published that most of the degradation of p53 mediated by E6 occurs in the nucleus (Stewart et al., 2005), and since we show that E6 dimerization is exclusively cytoplasmic, we think the two processes can coexist also in physiological conditions, i.e. in cells expressing p53.

So what to do with this paper?  The authors have done a lot of work here and much of it is clever.  But they evade and ignore these issues of overexpression and make unwarranted assumptions about the role of E6 interaction with scrib.   I think the work is publishable but only if the authors forthrightly confront these issues and couch the abstract, the entire discussion with the uncertainties that remain. 

We thank the Reviewer for commenting positively about the amount of work we did and its cleverness. Nevertheless, we agree with the Reviewer about her/his concern on possible overexpression issues. Thus, we have stressed throughout the whole manuscript that we employed an overexpression system (in the Abstract, Results, and Discusion sections). In addition, to take into account the Reviewer’s concern about uncertainties on the role of E6 interaction with hScrib, we added language in the Discussion section (page 17, lines 22-24 and lines 41-51) and mentioned the interactomic study by White et al. in the context of our reasoning. We also emphasized that the phenomenon we describe in this manuscript likely has significance in the context of cancer cells (page 1, line 24; page 2, lines 29-30; page 13, line 15; page 17, line 3).

They are really sticking their necks out when it is unclear what the role of E6-scrib association is at all.  For example, loss of scrib has been shown to sensitize cells to apoptotic cell death through cell competition.  Why would E6 want to do that?  That makes no sense and is the opposite of what the authors are apparently thinking.

We believe there are many possible explanations for this. While reiterating that we think E6-hScrib interaction has likely significance in cancer rather than during infection, apoptosis driven by cell competition and extrusion was shown to be driven by p53 activation (Wagstaff et al., Nat Comm 2016). So why would E6 want to do that? In our opinion simply because E6 can do that, as apoptotic extrusion, which may arise as a consequence of hScrib loss, is blunted in cancer cells through the concomitant p53 inactivation that E6 exerts in the nucleus.

Fig. 1.  In this experiment, HaCat cells that over-express E6 and E7 from the CMV promoter are assessed for downstream effects upon YAP/TAZ responsive genes.  This demonstrates that overexpresssion can induce downstream effects, it does not show that they do this at E6 and E7 expression levels encountered in either episomal primary or cervical cancer cell lines.  This is a general flaw of overexpression in the study and a lack of comparison to either natural infection or cervical cancers.  In part B, there is no comparison to vector transfected HaCat cells which is a flaw as we do not know if either E6 or E7 alone induce anchorage independence ---- the assay does demonstrate dramatic synergy of E6 and E7 expressed together.  Similarly, in part C there is neither a negative vector control nor a positive control with an activated YAP/TAZ to compare to E6 and E7. The paper leads us to the conclusion that the induced genes are due to YAP/TAZ activities, but this is unproven.

We agree with the Reviewer that our overexpression system may not be suitable for direct comparison with cervical cancer cells. However, the take-home message of Fig. 1A is that both E6 and E7 are concomitantly required to fully activate YAP/TAZ-target gene transcription. We are not addressing the reader to compare the extent of gene transcription in our overexpression system with primary counterparts. In addition, gene expression levels of HaCaT cells overexpressing E6, E7, or both are intrinsically compared to vector-transfected cells, as specified in the figure legend. Therefore, our data show a relative comparison that we believe it demonstrates that both oncoproteins are required, independently of the absolute level of transcription in our system. Same consideration applies to Fig. 1B for spheroid formation. Although the assay, in the way it is performed, determines stemness potential rather than anchorage independence per se, the comparison to vector-transfected cells would not be informative as HaCaT cells overexpressing either E6 or E7 are not capable to form spheroids, and parental HaCaT cells do not form spheroids as they are not endowed with stem-like potential (Tyagi et al., Sci Rep 2015). Regarding Fig. 1C, although the Reviewer is correct when stating that there is no formal demonstration that the upregulated basal markers are YAP/TAZ-mediated, a huge body of evidence exists in the literature that already formally demonstrated that YAP/TAZ activities determines stemness traits in most, if not all, organs, including the skin (Elbediwy et al., Bioessays 2016; Totaro et al., Nat Comm 2017; Heng et al., Front Cell Dev Biol 2020; Yuan et al., Nat Comm 2020). We think it is out of the scope of this manuscript to (re)demonstrate that the upregulation of epithelial basal markers is due to YAP/TAZ.

Fig. 2.  In part A, it is notable that C33A cells, although p53 mutated, have a p53 mutation that is degraded  by 16E6, unlike most p53 mutants  (published by Peter Howley's lab).  It is notable that both pdz and F47R mutations (defective for degrading p53) both ablate the stabilization of YAP at confluency.  Once again, these are overexpression experiments that show that E6 can do this, and not that it does do this under more physiologic conditions.  Just how much E6 is being made in these cells?  A comparison to E6 expression levels from primary keratinocytes harboring HPV16 genomes should be shown.

We again agree with the Reviewer saying that in our overexpression system there is surely more E6 than in primary counterparts endogenously expressing the viral oncoprotein. However, the take-home message was not to demonstrate that E6 upregulates YAP, because this was already proven in primary cervical cancer specimens (He et al., EMBO Mol Med 2015). We wished to determine more in detail the mechanism that leads to YAP upregulation (and here for the first time we demonstrate this also for TAZ). The observation that both E6 Y43E/F47R and E6 T149D/L151A do not upregulate YAP/TAZ independently of their overexpression is, in our opinion, sufficient to support the conclusions.

In part C, overexpression of E6 proteins to observe E6 dimerization through amino-terminal  E6 domaine interaction.  Transient overexpression of  Luc fused E6 together with YFP fused E6 results in a signal that is ablated by mutation of F47.  The dimerization of E6 through amino terminal interaction had been previously demonstrated in vitro, and results in the in vitro aggregation of E6 preparations upon concentration of the samples.  This is the first evidence for dimerization in vivo that I know of, but there is a problem in that F47 is required for the association of E6 with p53, and thus E6 could be multimerizing within HEK 293 cells vial association with p53 tetramers.  If the assay was performed in p53 null cells there would be more confidence.

We agree with the Reviewer, who is absolutely correct saying that (ideally) multimerization events of monomeric YFP-E6 and RLuc-E6 proteins binding p53 tetramers may lead to energy transfer between the E6 proteins, thus introducing a bias, and that Y43E/F47R substitutions may lead to reduced BRET due to the disruption of YFP-E6/p53 and RLuc-E6/p53 interactions rather than disruption of YFP-E6/RLuc-E6 dimers. However, it is known that efficient energy transfer for BRET measurements occurs at distances < 10 nm (Kobayashi et al., Nat Protocol 2019). A p53 tetramer has an average radius of 9 nm, with the two closest monomers at an approximate distance of 7 nm (Kitayner et al., Mol Cell 2006; PDB: 2ac0). E6 has an average radius of 3 nm (Zanier et al., Science 2013; PDB: 4giz). Therefore, YFP and RLuc moieties fused to E6 are most likely too far to efficiently transfer energy through tetrameric p53. Although we cannot exclude the occurrence of this phenomenon, we strongly believe that it provides a negligible contribution (if any), and BRET signals generated by monomeric YFP-E6 and RLuc-E6 bound to p53 monomers in the p53 tetramer are likely below the level of detection. Moreover, BRET signals measured through the expression of E6N fragments can be exclusively driven by E6 self-association, since E6N does not bind p53 (Fig. 2E), and PLA experiments in Fig. 3A have been performed in p53-null cells, which reinforce BRET data in HEK 293T cells expressing also p53.

In part C E6N associates with itself and that is lost upon mutation of F47 while E6N and p53 association are not observed in part E.  Again, such overexpression assays show that this self-association can occur in overexpression but does not show that it happens in physiologic conditions.    Indeed on lines 22-25 in results the authors state "These data thus indicated that E6 exists as a dimer in cells, and E6 homodimerization is indeed driven by the same interacting surface required for p53 binding. Therefore, our results pointed to a scenario in which, in parallel to PDZ-protein degradation, E6 could sustain YAP/TAZ induction through E6 homodimerization."  Actually that statement would be more accurate if they said "E6 can exist as a dimer in overexpression conditions using a surface that overlaps with the surface required for p53 interaction." 

We softened our claim as suggested by the Reviewer (page 9, line 49). We would like to underline however that such a difficult-to-study self-interaction requires overexpression systems to be readily measured and studied. We believe our work provides the proof-of-concept that E6 can indeed dimerize also in cells and evidence about its functional role. We agree with the Reviewer however about the necessity to address future works to precisely understand when and how E6 dimerization occurs also in vivo. As mentioned in one of our previous comments, we added language in the Discussion section speculating on this (page 17, lines 41-51).

Fig 3.  The PLA data in part A lacks positive and especially negative controls.  In part B the data may reach significance but it is not exactly compelling.  There are not other mutants included that all show no difference and the differences that are shown are small.  It may be significant, but is it important?  

Negative controls of PLA experiments of Fig. 3A were already provided in Fig. S3B. We included language in the main figure legend to direct the readers to the relative controls. A positive control in this case is misleading as it would require a self-interaction of a viral protein that behaves similarly to E6 in order to provide a useful contextualized information. We believe that Fig. 3B is important because it reinforces the evidence that E6 dimers have cytoplasmic localization, since living cells overexpressing YFP-E6 Y43E/F47R have a cytoplasmic fluorescence slightly lower than that of YFP-E6-expressing cells.

Reviewer 3 Report

Review of Messa et al.

The dimeric form of HPV16 E6 is crucial to drive YAP/TAZ up-regulation through the targeting of hScrib

This manuscript, describing the mechanism by which the E6 protein of HPV 16 regulates Yap/Taz transcription, is excellent.  The discovery that dimerization of E6 is required for this regulation is an important advance in the field and will be of interest to many.  The experiments are generally well executed, and the manuscript is very well written.

Specific comments:

Please provide units for vertical axes of all Western blots and graphs.

Page 1, line 30:  "and an increasing fraction of . . . oral cancers" is slightly incorrect.  The increase is in oropharyngeal cancers, which are throat cancers – not oral.  Please replace "oral" with "throat" or "oropharyngeal."

Page 1, lines 43-46:  This sentence is a run-on.  Please clarify by creating more than one sentence or adding non-comma punctuation.

Figure 1: A) It would be helpful to see quantitative data for the expression of the genes involved, for example a plot of the genes' expression levels relative to a control gene, in a Supplementary Figure.

Page 2, line 51:  Please specify the source of the HPV16 E6 sequence.

Page 8, line 20:  Recommend deleting "It has been recently demonstrated that."  That point has already been made and doesn't warrant repetition.

Page 8, Section 3.2:  Do the mutations used affect E6-AP binding?

Page 9, line 10:  Please describe the BRET assay briefly.

Figure 2:  D)  Please define the X axis label (Y/R) and provide units

Page 10, line 26 (also Figure 3 Legend title):  "and takes part in the E6/hScrib complex"

Page 10, line 30:  Please describe the Proximity Ligation Assay briefly. 

Page 10, line 29:  "E6 dimers in the cell"

Page 10, line 38:  "E6 dimers and E6 self-interaction" – please clarify how these are different. 

Figures 3 and 5:  The IP Westerns should include whole-cell lysates to show what fraction of the total is being precipitated.

Figure 3, A)  Please explain the 45° and semicircular arrow, shown between the images, in the legend.  (For example:  Both images show the same cell.  The bottom image has been rotated 45 degrees around the horizontal axis between the images.)  Should there be additional 45 degree arrows between the image on the right and the two on the left?

Figure 3, D) Please label the lanes.  Also, the E6 signal is very faint, hard to interpret.  Could anti-HA antibody be used instead? (The materials and methods indicate that anti-E6 was used for all Westerns.)

Figure 4, D) Please explain the 45° and semicircular arrow, shown between the images, in the legend.  Should there be an additional 90 degree arrow between the image on the left and the middle image?

Figure 4, lines 23-24:  Please clarify the difference between "colocalization" and "juxtaposition."

Page 16, line 39:  Commas before and after "in the cell" would clarify the sentence.

Author Response

Reviewer #3:  

This manuscript, describing the mechanism by which the E6 protein of HPV 16 regulates Yap/Taz transcription, is excellent.  The discovery that dimerization of E6 is required for this regulation is an important advance in the field and will be of interest to many.  The experiments are generally well executed, and the manuscript is very well written.

We are grateful to the Reviewer for judging our study positively and finding our manuscript excellent and our work “an important advancement in the field” and “of interest to many”.

Please provide units for vertical axes of all Western blots and graphs.

We modified all the figures as requested.

Page 1, line 30:  "and an increasing fraction of . . . oral cancers" is slightly incorrect.  The increase is in oropharyngeal cancers, which are throat cancers – not oral.  Please replace "oral" with "throat" or "oropharyngeal."

We modified the sentence as suggested (page 1, line 40; page and line numbers refer to the cleaned manuscript with hidden track changes).

Page 1, lines 43-46:  This sentence is a run-on.  Please clarify by creating more than one sentence or adding non-comma punctuation.

We also modified this sentence, as suggested (page 2, lines 10-11).

Figure 1: A) It would be helpful to see quantitative data for the expression of the genes involved, for example a plot of the genes' expression levels relative to a control gene, in a Supplementary Figure.

We included non-normalized quantitative gene expression profiles in the new Fig. S1B, as requested by the Reviewer. For sake of clarity, we plotted the mean log2 fold-change values for each gene of each cell line under study (HaCaT cells expressing HPV16 E6, E7, or both). We did not plot the expression values for empty vector-transfected cells as they represent the control sample against which the analysis is performed, and therefore they are intrinsically set as 0. However, all genes’ expression levels are internally compared to a control gene (GAPDH) for each sample, so the comparison to a control gene was already implicit. For completeness, although the Reviewer did not request it, we also provided non-normalized log2 fold-change values for gene expression profiles shown in Fig. 5C and included in the new Fig. S5C.

Page 2, line 51:  Please specify the source of the HPV16 E6 sequence.

We specified the information as requested (page 3, line 15).

Page 8, line 20:  Recommend deleting "It has been recently demonstrated that."  That point has already been made and doesn't warrant repetition.

We modified the sentence as suggested (page 8, line 17).

Page 8, Section 3.2:  Do the mutations used affect E6-AP binding?

Considering the crystal structure of the E6/E6AP/p53 complex (Martinez-Zapien et al., Nature 2016) and NMR model of dimeric E6 (Zanier et al., Structure 2012), E6AP binding is most likely not affected by the amino acid substitutions we introduced on E6 throughout this work. In fact, the E6/E6AP interaction involves a different region on E6 protein surface that does not include the N-terminal alpha-helix driving both E6 homodimerization and E6/p53 binding, nor the amino acids required to interact with PDZ proteins. Therefore, based on the structural information mentioned above, both E6 Y43E/F47R and E6 T149D/L151A should still normally bind to E6AP as wild-type E6.

Page 9, line 10:  Please describe the BRET assay briefly.

We added language in the Results section (page 9, lines 25-27) as requested.

Figure 2:  D)  Please define the X axis label (Y/R) and provide units

We modified the figure as requested.

Page 10, line 26 (also Figure 3 Legend title):  "and takes part in the E6/hScrib complex"

We modified the sentence as indicated (page 11, line 1).

Page 10, line 30:  Please describe the Proximity Ligation Assay briefly.

We added language in the Results section (page 11, lines 6-8) as requested.

Page 10, line 29:  "E6 dimers in the cell"

We modified the sentence as indicated (page 11, line 4).

Page 10, line 38:  "E6 dimers and E6 self-interaction" – please clarify how these are different.

We agree with the Reviewer that the original sentence was confusing, as there is no difference. Thus, we modified the sentence to make it clearer (page 11, lines 15-16).

Figures 3 and 5:  The IP Westerns should include whole-cell lysates to show what fraction of the total is being precipitated.

Whole-cell input controls for the anti-hScrib IP shown in Fig. 3C were already provided in Fig. S3C. We apologize for not including input controls for the anti-hScrib IP shown in Fig. 5A, which have been now included in new Fig. S5A. For sake of space in the main figures, we showed whole-cell extract input controls in supplementary figures, but we now included language in the respective main figure legends to direct the readers to the relative input controls.

Figure 3, A)  Please explain the 45° and semicircular arrow, shown between the images, in the legend.  (For example:  Both images show the same cell.  The bottom image has been rotated 45 degrees around the horizontal axis between the images.)  Should there be additional 45 degree arrows between the image on the right and the two on the left?

We added language in the legend of Figure 3 explaining the semicircular arrow and added a second semicircular arrow to depict the rotation of the cell for the 3D reconstruction as suggested by the Reviewer.

Figure 3, D) Please label the lanes.  Also, the E6 signal is very faint, hard to interpret.  Could anti-HA antibody be used instead? (The materials and methods indicate that anti-E6 was used for all Westerns.)

We would like to clarify that untagged E6 proteins were always blotted using anti-E6 primary antibodies (as those shown in Fig. 2A), while tagged E6 proteins (including HA-tagged versions) were blotted using primary antibodies directed against the tag. Thus, the blot shown in Fig. 3D was obtained using an anti-HA primary antibody. We thank the Reviewer for raising this point and added language in the Materials and Methods section to clarify this detail (page 5, lines 47-51). We retrieved the original membrane where we blotted HA-E6 proteins shown in Fig. 3D and tried to re-blot them with the same anti-HA antibody used originally, but unfortunately we failed in improving the result. However, as mentioned in the figure legend, those faint bands are due to the scarcity of the material used for the Western blot, as it derives directly from the samples that were subjected to BRET measurements in the 96-well plate (typically a triplicate, containing no more than 100/150K cells in total).

Figure 4, D) Please explain the 45° and semicircular arrow, shown between the images, in the legend.  Should there be an additional 90 degree arrow between the image on the left and the middle image?

As suggested by the Reviewer, we added language in the legend of Fig. 4D explaining the semicircular arrow and added a second 90° arrow to depict the rotation of the cell for the 3D reconstruction.

Figure 4, lines 23-24:  Please clarify the difference between "colocalization" and "juxtaposition."

We modified the sentence as requested, explaining that “colocalizatioon” refers to the presence of both signals in the same pixels, while “juxtaposition” refers to the presence of one signal in close proximity to the other

Page 16, line 39:  Commas before and after "in the cell" would clarify the sentence.

We added commas as suggested (page 17, line 52).

Round 2

Reviewer 2 Report

The authors have properly revised text in the abstract, results and discussion to draw attention to the fact that most of the data is derived from experiments where E6 is  overexpressed.  This will allow readers to better appreciate the nature of the experiments that were performed and the proper conclusions to be drawn, and the questions that remain.  This benefits both the authors and the readers and I commend the authors for doing this directly, clearly, completely, and without evasion.  

The authors have fully addressed my concerns.